# Knowledge, attitudes and practices on household solid waste management and associated factors in Gelemso town, Ethiopia

Hailu Eshete[1], Asnake Desalegn[2], Fitsum Tigu🔘[2]*

1 Department of General Biology, College of Natural and Computational Sciences, Addis Ababa University, Addis Ababa, Ethiopia, 2 Department of Microbial, Cellular and Molecular Biology, College of Natural and Computational Sciences, Addis Ababa University, Addis Ababa, Ethiopia

* fitsum.tigu@aau.edu.et

**Data Availability Statement:** All relevant data are within the paper and its Supporting Information files.

## Abstract

Improper solid waste management in urban and semi-urban cities of developing countries is the source of environmental pollution and public health concern. This study aimed to assess the households' knowledge, attitudes and practices of solid waste management and associated factors in Gelemso town. A community-based cross-sectional study design was used to assess the households' KAP of SWM in Gelemso town. A systematic random sampling technique was used to recruit 390 households from Gelemso town and data was collected using a structured questionnaire. Data analysis was carried out using SPSS version 22.0 software. Bivariate and multivariate analyses were used to predict the improper SWM practices. From 390 households included in the study, 61.3% of them were females. Generally, most households had correct knowledge and positive attitudes towards SWM but poor practice was observed in the study area. About 96% of the households considered solid waste as a source of environmental pollution and close to 92% of them replied that solid waste can be used for compost preparation. Majority (87.4%) of the households "strongly agreed" about the potential risk associated with improper solid waste disposal and nearly 80% of them also "strongly agreed" that proper SWM is crucial to create a healthy environment in the community. Nearly 80% had practiced improper SWM. Logistic regression analyses indicated that lack of experience in sorting solid waste, ways of removal, knowledge about reduce, reuse and recycle, absence of adequate solid waste landfills, and lack of door-to-door waste collections services were identified as the major contributing factors for the improper SWM practice in Gelemso town. In conclusion, the majority of the households had practiced improper SWM, such as disposing of solid waste in the backyard, along the roadsides, in gully and burned. These malpractices can significantly affect the environment and public health of the residents.

## 1. Introduction

Solid waste is defined as unwanted and sometimes hazardous materials with low liquid content generated from municipality, industry and commerce as well as agriculture and other related

**Funding:** HE is an MSc student of Addis Ababa University (AAU), College of Natural and Computational Sciences. He obtained student financial support for the completion of MSc thesis from Graduate Programs of AAU. The support is only for data collection and processing. The funders had no role in study design, data collection and analysis, decision to publish, or preparation of the manuscript.

**Competing interests:** The authors have declared that no competing interests exist.

sectors [1]. However, in many unindustrialized countries including Ethiopia, the main sources of solid wastes are generated from households and agriculture [2]. Solid waste management (SWM) is a complex activity that involves the collection, transportation, processing and disposal of solid wastes in a way that is best addressed for human welfare [3]. Failure of proper SWM in the urban and semi-urban areas result in proliferation of solid waste associated diseases [4] and provide suitable breeding sites for vectors like mosquitoes, flies and rodents which in turn are responsible for public health problems [5].

Proper solid waste disposal is a big challenge for most developing countries including Ethiopia. Studies indicated that, households' perception, awareness, knowledge, attitudes, and behavior towards solid waste had significant impact on SWM [6, 7]. In addition, the disproportionate generation of solid waste and SWM systems of the municipalities become problematic for many countries [8]. Moreover, such issues are associated with various factors including lack of adequate infrastructures, lack of legal enforcement on environmental issues, irregular and unplanned dumping of solid wastes, population increase, urbanization due to rural-urban migration, insufficient capital to run SWM process and absence of new technology for waste disposal [9].

In Ethiopia over 67.4% of generated solid waste is dominated by organic biodegradable materials which can be recycled [2], however, only 5% is recycled in non-standard ways [4]. The majority of the solid waste generated by the municipality are either illegally disposed of without pre-treatment or traditionally burned in the open air [2]. Hence proper SWM requires the commitment by the municipality and ensuring active involvement of the community. Not only providing solid waste infrastructure, but also in depth understanding of the community's perception, knowledge and behavior are quite important [10]. A study reported that, there was a gap between public perception and awareness, as it identified in the study [11]. Moreover, poor awareness and delay of collection fees payment to the micro and small enterprises (MSEs) [12, 13] and irregular solid waste collection and exclusion of poor residences in the solid waste collection systems are other factors that hindered the process of proper SWM [2]. Besides knowledge and awareness, attitude is an important element that shapes the society to understand their surroundings, keep the environment healthy and practice a proper SWM.

Various studies have been conducted in Ethiopia on assessment of factors influencing these behaviors including knowledge, attitudes and practices (KAP); however no study has been conducted in Gelemso town regarding the household KAP towards SWM. The town has limited SWM facility, most of the time the solid waste is indiscriminately disposed along roadsides and into open areas and hence the current status of the solid waste problem in the town has reached a critical stage. Thus, the aim of this cross-sectional study was to assess the households' KAP towards SWM and the factors associated with improper SWM in the town. The findings of the current study may help the local government to fill the gap on SWM. It could also serve as baseline information for future study.

## 2. Methods

### 2.1. Description of the study area

Gelemso is one of the administrative towns in West Hararghe zone of Oromia National Regional State and the origin of Khat. It is located 376 km away from Addis Ababa, the capital city of Ethiopia and 70 km from Chiro, zonal capital city. It has a total area of 46,000 hectare and administratively divided into two *Kebeles* (lowest administrative units in Ethiopia). The town is largely characterized by highland climate conditions. According to the Gelemso municipal office report (2017), the total population of Gelemso is 43,837 (23,735 males and 20,102 females). The major economic activities of the town are trade (mainly of Khat), urban

agriculture, micro and small-scale enterprises and other informal businesses. The town also consists of a zonal hospital, four health clinics, technical and vocational institutes, preparatory and high schools, four elementary schools and three kindergartens with several government and private sectors.

The city municipality of Gelemso has waste management section, which is responsible for waste management system of the city government. Under which one Micro and Small Enterprise (MSE), comprising of 10 members, one waste collection service vehicle and two shared containers are allocated for this purpose. The municipality has no official damping site in or around the city. Previously the collected wastes were damped to an open-pit (created during rock extraction) near the city. Recently, the collected wastes are damped to an open area in the boarder of the city. The MSE members are responsible for picking waste from home-to-home, and shared container and roadsides to the final damping site, however, the MSE collect wastes only from registered and households who are willing to pay monthly service charges. The municipality has partial subsidy to MSE. Thus, open waste damping systems are commonly practiced by the city dwellers.

## 2.2. Study design and sample size determination

A community-based cross-sectional study design was used to assess the households' KAP on SWM in the Gelemso town from March to June 2020. Sample size was determined by assuming that half of the respondents had knowledge on SWM with estimated prevalence rate of 50% (p = 0.5) at 95% confidence level (Zα/2 = 1.96) and 5% of marginal error (d = 0.05) [14]. Based on the formula, 422 participants were involved including the 10% non-response rate. The researcher obtained total list of names and number of households (5144) from Gelemso municipality office. Since sample size to study population is greater than 0.05, the correction formula of (n/1+n/N) was used to determine the final sample size, 396 (422/1+422/5144) plus 1.5% non-response rate. Next to this, the value of k was calculated by 5144/396 = 13 which is approximately 13. Finally, a systematic random sampling technique was used to select households from the two *Kebeles*. Then the first household was located by lottery method and the next household was selected by every 13th (kth) from the total lists of the households. Furthermore, two *Kebele* leaders, two municipality officials and six members from MSE were purposely included in the study.

## 2.3. Operational definition

**Improper solid waste management**: is defined as a kind of illegal solid waste management practice in which the unsorted solid wastes and/or the generated household solid wastes were damping at legally unauthorized place; **Landfill**: is a facility designed by the municipality for the solid wastes disposal.

## 2.4. Data collection

**2.4.1. KAP survey.** Participatory mixed approaches (both qualitative and quantitative procedures) were used to collect data from the households. Quantitative data were collected through interview using a structured pretested questionnaire. The questionnaire consists of both open and closed ended questions about socio-demographics, households' knowledge, attitudes and practices towards SWM. The questionnaire was adopted from different guidelines and published articles with some modification [8, 15, 16]. First, the questionnaire was prepared in English and translated to Afan Oromo (local language). Field observation was conducted in order to understand the actual households' practice towards SWM such as open dumping sites in river sides, valley, road, and waste collection and transportation systems as

well as waste disposal facilities in the town. Photographs showing the solid waste management practices and disposal sites in the community were taken by the principal investigator during the field visit.

**2.4.2. Key informant interview.** A total of 10 key informants consisting of two *Kebele* leaders, two municipal office experts and six members of MSEs were interviewed for the validation of the interviewee responses in the town.

**2.4.3. Focus group discussion.** A group of 10–12 participants were drawn for focus group discussions (FGDs) among the elderly and local community representatives in Gelemso town to understand the KAP responses of the community towards SWM. A total of two focus group discussions were conducted in each *Kebele* during March and May 2020, respectively.

## 2.5. Data analysis

Completeness and consistency of the data were checked and double entered into SPSS version 22.0 software. Descriptive statistics were done to summarize the socio-demographics and KAP data. Bivariate analysis was used to determine the association between the dependent and independent variables. Logistic regression analysis was used to determine the best predictors of improper solid waste management practices.

## 2.6. Ethical statement

Ethical approval was obtained from Addis Ababa University, College of Natural and Computational Sciences, Institutional Review Board (CNCS-IRB) with reference number: IRB/040/ 2020. Before data collection, all study participants were informed about the purpose of the study and the rights to participate or decline to participate in the study. The respondents were also informed about the confidentiality of their information and assured that all the data will be used for research purposes. Additionally, before collecting the data through the interview, all participants signed a written consent form. The study included adult participants of age $\geq$ 18 years old.

# 3. Results

## 3.1. Socio-demographic characteristics

A total of 390 households (61.3% female and 38.7% male) were involved in the study with a response rate of 98.5%. Among which 66.9% lived in the city over 10 years. Close to 50% of the participants earned an average monthly income of 75 USD and over 60% of the participants were self-employed (Table 1).

## 3.2. Households' knowledge and attitudes towards SWM

About 96% of the households gave the correct response regarding solid waste as sources of pollutant for the environment and 88.2% of them were associated with diarrhea, typhoid, and cholera (Table 2). Similarly, 78.2% of the household knew the recyclable items from the solid waste and almost similar figure of household sorted and sold these items to recycling companies. Over 87% of the household claimed that burning solid wastes causes health risks to the community. Nearly 90% of the respondents also knew that improper solid waste can eventually lead to pollution of rivers, lakes and wells. Likewise, close to 92% of the household also believed that solid wastes can be used for compost preparation. A relatively less knowledge related response of (60.8%) was recorded on how the solid waste had been used for making wealth in a household.

**Table 1. Socio-demographic characteristic of the respondents.**

| Variable | Category | Frequency (%) |
|---|---|---|
| *Kebele* | 01 | 231 (59.2) |
| | 02 | 159 (40.8) |
| Sex | Male | 151 (38.7) |
| | Female | 239 (61.3) |
| Age (year) | 18–30 | 87 (22.3) |
| | 31–45 | 180 (46.2) |
| | 46–60 | 103 (26.4) |
| | > 60 | 20 (5.1) |
| Education level | No education | 45 (11.5) |
| | Primary school | 102 (26.2) |
| | Secondary school | 90 (23.1) |
| | Diploma | 86 (22.0) |
| | Bachelor' degree & above | 67 (17.2) |
| Residence period (year) | < 2 | 26 (6.7) |
| | 2–5 | 56 (14.4) |
| | 6–10 | 47 (12.1) |
| | > 10 | 261 (66.9) |
| Family size | 1–3 | 121 (31.0) |
| | 4–6 | 167 (42.8) |
| | 7–9 | 75 (19.2) |
| | > 10 | 27 (6.9) |
| Monthly income (USD) | < 35 | 67 (17.1) |
| | 35–115 | 186 (47.7) |
| | > 115 | 137 (35.1) |
| Occupation | Self-employed | 235 (60.3) |
| | Government employed | 113 (29.0) |
| | Company employed | 32 (8.2) |
| | Others | 10 (2.6) |

The households' attitudes towards the SWM among the Gelemso community were assessed and summarized in Table 3. The result indicated that the majority (87.4%) of the households "strongly agreed" that solid waste is one of the environmental problems of the city which

**Table 2. Households'response to knowledge questions on SWM.**

| Knowledge question | Frequency (%) | |
|---|---|---|
| | Yes | No |
| Is solid waste source of pollution for the environment? | 373 (95.6) | 17 (4.4) |
| Burning of solid wastes causes health risks (eg. bronchitis and asthma) | 339 (86.9) | 51 (13.1) |
| Waste papers, plastic bags, a piece of metal, wood and cloths are recyclable? | 305 (78.2) | 85 (21.8) |
| Do you consider solid waste as a wealth? | 237 (60.8) | 153 (39.2) |
| Can solid waste be sorted and sold for recycling companies. | 310 (79.5) | 80 (20.5) |
| Compost or organic fertilizer can be prepared from solid waste? | 358 (91.8) | 32 (8.2) |
| The amount of solid waste can be reduced by reusing at household level. | 332 (85.1) | 58 (14.9) |
| Illegal damping of solid waste causes diarrhea, typhoid, and cholera. | 344 (88.2) | 46 (11.8) |
| Sorting of solid waste at home level helps for SWM. | 287 (73.6) | 103 (26.4) |
| Improper dumping of solid wastes can eventually lead to pollution of rivers, lakes and wells. | 350 (89.7) | 40 (10.3) |

Table 3. Households'attitudes towards SWM.

| Attitude question | Frequency (%) | | | |
|---|---|---|---|---|
| | Strongly agree | Agree | Disagree | Strongly disagree |
| Solid waste is anything without value | 41 (10.5) | 71 (18.2) | 102 (26.2) | 176 (45.1) |
| Solid waste is one of the environmental problems that need an immediate attention | 341 (87.4) | 24 (6.2) | 19 (4.9) | 6 (1.5) |
| Solid wastes can be reduced, reused and recycled (3Rs) | 274 (70.3) | 57 (14.6) | 52 (13.3) | 7 (1.8) |
| Every household should have responsibility for the proper collection and disposal of solid wastes | 242 (62.1) | 111 (28.5) | 26 (6.7) | 11 (2.8) |
| Proper solid waste disposal is the responsibility of everyone | 210 (53.8) | 136 (34.9) | 33 (8.5) | 11 (2.8) |
| Proper SWM is important for creating healthy environment | 307 (78.7) | 50 (12.8) | 22 (5.6) | 11 (2.8) |
| SWM is a burning issue in the town | 266 (68.2) | 74 (19.0) | 26 (6.7) | 24 (6.2) |
| The city government should conduct regular supervision and control on illegal dumping of solid waste in the town | 21 (5.4) | 44 (11.3) | 226 (57.9) | 99 (25.4) |
| Selling plastic waste for recycling is the best way to manage solid wastes | 212 (54.4) | 114 (29.2) | 49 (12.6) | 15 (3.8) |

required an immediate attention by the city government. Over 70% of them accepted the principle that solid waste can be reduced, reused and recycled. About 79% of households "strongly agreed" that proper SWM is important to create a healthy environment among the community. However, most households pointed out that the city government did not create enough awareness on SWM and control on illegal solid waste disposal in the city. Likewise, about 68% of the households "strongly agreed" that SWM was a burning issue in the town. Slightly over 62% and 28.5% of the households "strongly agreed" and "agreed" that proper solid waste collection and disposal was the responsibility of every household, respectively. Over half (54.4%) of the households "strongly agreed" and 29.2% of them "agreed" that selling plastic waste for recycling is the best way to manage solid wastes.

### 3.3. Solid waste removal methods

Slightly over 45% of the households separated their solid waste at household level before disposal. However, the majority (79.4%) of the households disposed of their solid waste illegally, either in the backyard with sacs (57.9%) or along the roadsides and in gully (21.5) (Table 4). More than half (57.4%) of the households had practiced 3Rs strategies (Fig 1) and the remaining households (42.6%) burned the solid waste in their compound as indicated in Table 4.

The majority (89.2%) of the households reported the absence of adequate solid waste landfills or damping sites in the city and thus had practiced an improper SWM (Table 4). As indicated in Fig 2, slightly over 82% of the households mentioned that the current status of solid waste has increased from time to time in terms of volume as well as compositions in the city. Besides the absence of solid waste landfills, 70% of the households have no access to door-to-door solid waste collection services as mentioned in Table 4 and over 60% of the interviewed households and key informants reported the poor solid waste collection and disposal service of the city government (Fig 3). Mainly solid waste is transported to the nearest container by private waste collectors, which account about 51%, and followed by household members and MSE.

### 3.4. Factors affecting proper SWM in Gelemso town

In order to determine factors contributing to improper SWM, independent variables having p-value < 0.05 in bivariate analysis was used for fitting the logistic regression model. Table 5

**Table 4. Households' practices on SWM.**

| Practice question | Frequency (%) |
|---|---|
| *Do you separate solid wastes before disposal*? | |
| Yes | 178 (45.6) |
| No | 212 (54.4) |
| *How do you get rid of solid wastes from home*? | |
| Dumped in the backyard with sacs | 226 (57.9) |
| Dumped along roadsides or in gully | 84 (21.5) |
| Dumped in disposal site | 31 (7.9) |
| Buried in the soil | 49 (12.6) |
| *Do you practice reduce, reuse and recycle strategy for SW or not use 3R*? | |
| Reduce by compost preparation (R-1) | 43 (11.0) |
| Reuse (R-2) | 107 (27.4) |
| Selling for recycling business (*Korale* and *Lewache**) (3R) | 74 (19.0) |
| Burn (Not use 3R) | 166 (42.6) |
| *Are there adequate solid waste landfills or dumping sites in the town*? | |
| Yes | 42 (10.8) |
| No | 348 (89.2) |
| *Do you have access to door-to-door waste collection service*? | |
| Yes | 117 (30.0) |
| No | 273 (70.0) |
| *How do you transport solid waste to the nearby container*? | |
| By household members | 70 (17.9) |
| By MSE with fair payment | 70 (17.9) |
| By private waste collectors | 198 (50.8) |
| Other means | 52 (13.3) |

* *Korale*: informal waste collectors mainly collect recyclable materials (plastics, glasses, corrugated iron sheets, tins and car batteries) from door-to-door, in the streets, scavengers on the dumpsite, wholesalers and waste resellers. * *Lewache*: informal waste exchangers, they collect certain waste (textiles and shoes etc...) from door to door in exchange with new plastic containers, sauce pans, spoons, and other household items. Both *Korale* and *Lewache* are selling their collected waste to middlemen, who in turn sell them to brokers of recycling companies.

indicated that those households who had no experience of sorting solid waste, lacking the right solid waste disposal method and fundamental principles of 3R were more likely to practice improper SWM than those who had the required experience, method and principles of SWM. Similarly, households who had no access to solid waste landfills, and door-to-door waste collection services were more likely to practice improper SWM than those who had the services with AOR = 0.084, 95% CI = 0.024–0.296, $p < 0.001$ and AOR = 0.105, 95% CI = 0.039–0.283, $p < 0.001$, respectively.

## 4. Discussion

Various studies indicated that demographic characteristics of the household comprising sex, age, level of education, family size, income and occupation are important variables to understand the KAP of SWM [17, 18]. In this study, most of the household had 4–6 persons and earning less than 50 USD per month. In the study of Limon and Villarino, 2020 [16], which was conducted on KAP of food waste, family size and income of the household, were identified as the significant factors for the disposals of food waste. Most households had an age group of 31–45 years old, they had a family with 4 to 6 members and over 60% of them were women.

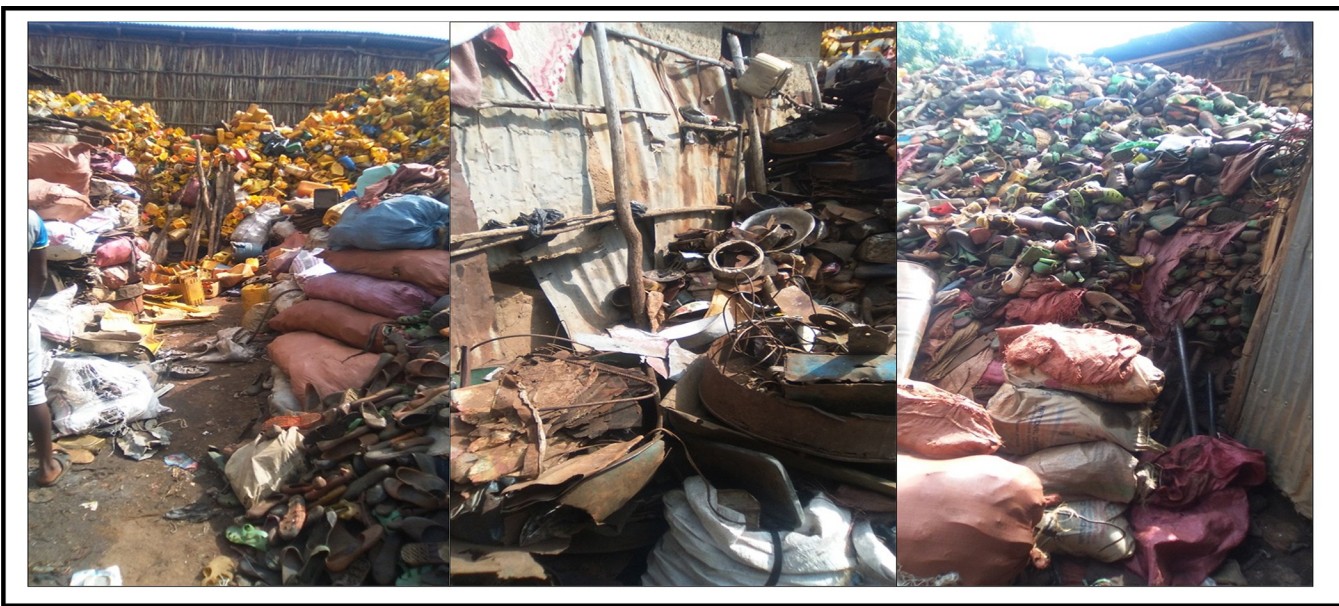

**Fig 1. Collection of solid wastes by *Korale* and *Lewache* for recycling (Source: Field observation by researcher).** Left to right: plastics, beverage cans, glass and bottles; metals, corrugated iron sheets; and textiles and shoes.

This observation was in line with the studies conducted in elsewhere [16, 19]. The variations in the demographic characteristics of the households could be due to the socioeconomic and cultural differences among different settings.

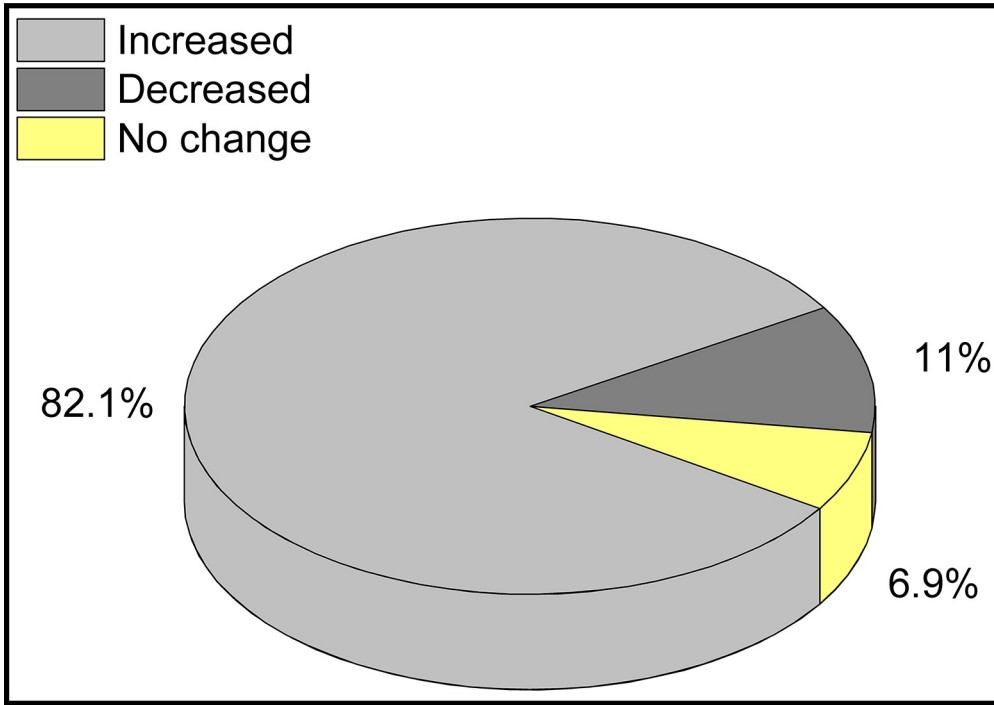

**Fig 2. Multiple responses of households showing the current status of solid waste volume and composition in Gelemso town.**

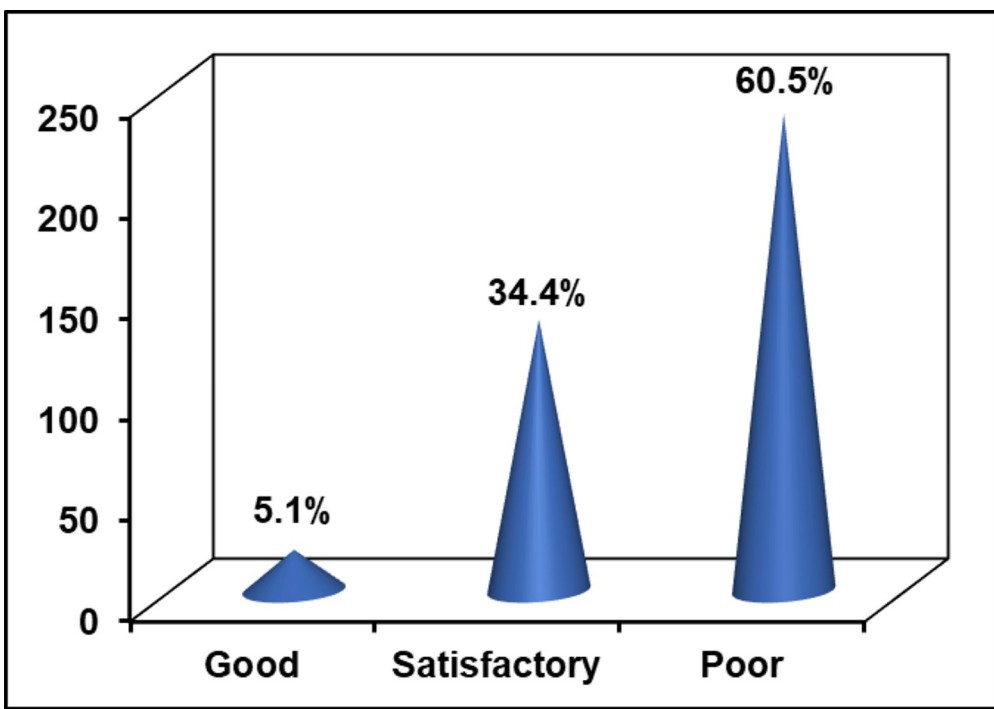

**Fig 3. Household responses towards city government solid waste collection and disposal services.**

The finding of this research indicated that households had a high level of knowledge on solid waste, and SWM as well as positive attitudes towards SWM. However, most of the households had practiced improper solid waste disposal. Likewise, Lema et al. [5] also reported that over 83% of the Asella town in Ethiopia had the same improper solid waste disposal system. This might be associated with the absence of landfill and door-to-door waste collection services in the two places. Logistic regression analysis showed that there are statistically significant differences among households who had access to solid waste landfills and door-to-door waste collection services than those who had no services ($p < 0.05$).

The present study showed that about 90% of the households correctly understood the adverse effects of improper solid waste disposal on the environment. Based on their

**Table 5. Bivariate and multivariate analyses of factors associated with SWM practice in Gelemso town.**

| Variables | SWM Practice | | | OR (95% CI) | | | | P value |
|---|---|---|---|---|---|---|---|---|
| | Category | No | % | COR | AOR | 95% CI | | |
| Experience of sorting solid waste before disposal | Yes | 178 | 45.3 | 1 | 1 | | | |
| | No | 212 | 54.4 | 0.099 | 0.077 | 0.023 | 0.261 | < 0.01 |
| Methods for get rid of solid waste from home | Yes | 80 | 20.5 | 1 | 1 | | | |
| | No | 310 | 79.5 | 0.163 | 0.142 | 0.063 | 0.323 | < 0.01 |
| Knowledge about the 3R strategy | Yes | 224 | 57.4 | 1 | 1 | | | |
| | No | 166 | 42.6 | 0.126 | 0.217 | 0.061 | 0.775 | 0.019 |
| Having adequate solid waste landfills/dumping sites | Yes | 42 | 10.8 | 1 | 1 | | | |
| | No | 348 | 89.2 | 0.161 | 0.084 | 0.024 | 0.296 | < 0.001 |
| Having access to door-to-door waste collection service | Yes | 117 | 30.0 | 1 | 1 | | | |
| | No | 273 | 70.0 | 0.146 | 0.105 | 0.039 | 0.283 | < 0.001 |

understanding, solid wastes were the sources of contaminants for the environment which leads to pollution of rivers, lakes and wells; it can be the sources of various diseases such as bronchitis, asthma, diarrhea, typhoid, cholera and other vector borne illness to humans. A study in Swaziland indicated that people who are living very close to the damping site are victims of malaria, chest pains, cholera, and diarrhea [4].

On the other hand, the households' awareness about the usefulness of the solid waste for making wealth via applying the 3Rs strategies (reduced, reused and recycled), and making compost as fertilizers were as high as 80%. This research finding is relatively comparable with the study reported in Zimbabwe [20], much higher than the research findings in Ethiopia [21] and other studies conducted elsewhere in the world [15, 22, 23]. Differences in the knowledge level of the respondents can be due to educational status, nature of solid waste, infrastructure conditions and the awareness levels across the studies.

In terms of attitudes toward SWM, higher levels of positive attitudes were observed among the Gelemso community. Accordingly, most of the households strongly agreed that solid waste is a source of contaminant for the environment that required immediate attention by the local government. Furthermore, the households had perceived a positive attitude to proper solid waste disposal; they also accepted the principles of 3R for the creation of a healthy environment for the society. This finding is in agreement with the previous studies conducted elsewhere [20, 22]. The positive attitude toward proper solid waste removal might be a good opportunity to avoid the adverse effects of solid waste in the environment and huge potential for the creation of jobs in the SWM systems such as micro-enterprise waste collection services, composting and other waste recycling activities.

Solid waste disposal is the major problem of most developing countries including Ethiopia. In this study, close to 80% of the households were disposed of their solid waste in illegal ways (backyard, along the roadsides, in gully and burned). Although the degree of which differs from place to place, improper solid waste disposal is a common challenge of most cities in Ethiopia [4, 8, 24, 25] and also other African countries [6, 26–28]. The current study revealed that those households who had no experience of sorting solid waste, lacks the right ways of solid waste removal and those households who did not practice 3Rs strategies were identified as the major contributing factors for improper disposal of solid waste at Gelemso city. Furthermore, the high improper solid waste disposal practices in this particular study may be associated with the absence of adequate solid waste landfills, damping sites and lack of access to door-to-door solid waste collection services in the city as ascertained from the respondents and direct field observation. Logistic regression analysis also revealed a direct association between improper SWM and lack of access to solid waste landfills and/or door-to-door waste collection services.

Higher illegal solid waste disposal was reported in this study, 79.5%, compared to other studies conducted in Ethiopia such Addis Ababa, 25% [29], Adama, 38% [30], Gondar, 69.7% [24], Bahir Dar and Debre Berhan, nearly 75% [25, 31]. And also from other African countries like Accra, Ghana, 39% [32], Kampala, Uganda, 59% [33], Owerri, Nigeria, 66.3% [34], and Keko Machungwa, Tanzania, 71.5% [28]. Possibly these variations can be explained by differences in infrastructure, urbanization, community level of understanding and intervention measures among others.

Besides inadequate door-to-door solid waste collection service, households' experience of solid waste separation, reduction, reuse and recycling were unsatisfactory. As logistic regression analysis indicated households who had no experience in sorting solid waste at household level had practiced improper SWM than those who had the experience (AOR = 0.077, 95% CI = 0.023–0.261, $p < 0.01$). Similarly, there was statistically significant ($p = 0.019$) variation in SWM practice between households who applied the 3R strategy and those who did not. In the

present study 55% of the households did not separate solid waste prior to land filling, which is much lower than the study conducted in Ghana, 82.7% [32], Uganda, 78.7% [33] and Iran, over 80% [22].

In Ethiopia, recycling of solid waste is mainly conducted by informal sectors and not well practiced by most regions. Various factors are associated with it such as inefficient organizational structure, absence of clear policy and regulation are among others [2]. In our study, very few households recycled their solid waste through *Korale* and *Lewache* [35, 36]. On the other hand, the majority of the solid waste that was generated in Gelemso town are biodegradable, only insignificant amounts of which are recycled through informal institutions [2]. Thus, launching of formal institutions to recycle and manage the solid wastes that are generated in every household is expected from the Gelemso municipality office.

## 5. Conclusion

The finding of this study indicated that the households' knowledge and attitudes on solid waste and SWM is high but the level of practice is very poor. Most households disposed of their solid waste either in the backyard with sacs or along the roadsides, in gully and burned in open air. Lack of experience in sorting solid waste, removal methods, knowledge about 3R, absence of adequate solid waste landfills, and lack of door-to-door waste collections services were the major contributing factors for the improper SWM practice in Gelemso. This can significantly affect the environment and public health of residents. Therefore, proper SWM should be advocated by the local government, and other stakeholders to keep the environment healthy for the local people. Furthermore, the municipality of Gelemso town needs to improve the SWM systems of the town through construction of adequate solid waste landfills, shared containers and door-to-door solid waste collection services to the residents of Gelemso.

## Supporting information

**S1 File. Survey questionnaire (English version).**
(DOCX)

**S2 File.**
(PDF)

## Acknowledgments

Authors gratefully acknowledge Addis Ababa University, School of Graduate Program for the provision of technical and administrative support to this project. Authors are also thankful to study participants, Gelemso Municipality Office, the elderly and local community representatives of Gelemso town for their valuable information and support.

## Author Contributions

**Conceptualization:** Hailu Eshete, Asnake Desalegn, Fitsum Tigu.

**Data curation:** Hailu Eshete.

**Formal analysis:** Hailu Eshete, Asnake Desalegn, Fitsum Tigu.

**Investigation:** Hailu Eshete, Asnake Desalegn.

**Methodology:** Hailu Eshete, Asnake Desalegn, Fitsum Tigu.

**Software:** Hailu Eshete, Asnake Desalegn, Fitsum Tigu.

**Supervision:** Asnake Desalegn.

**Writing – original draft:** Fitsum Tigu.

**Writing – review & editing:** Asnake Desalegn, Fitsum Tigu.

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
