## [Decision Letter · Decision Letter 0]

4 Apr 2022

PONE-D-22-02551Assessment of household experience of solid waste management in Gelemso town, EthiopiaPLOS ONE

Dear Dr. Yifat,

Thank you for submitting your manuscript to PLOS ONE. After careful consideration, we feel that it has merit but does not fully meet PLOS ONE’s publication criteria as it currently stands. Therefore, we invite you to submit a revised version of the manuscript that addresses the points raised during the review process.

We look forward to receiving your revised manuscript.

Kind regards,

Saqib Bashir

Academic Editor

PLOS ONE

Journal Requirements:

Reviewers' comments:

Reviewer's Responses to Questions

**Comments to the Author**

1. Is the manuscript technically sound, and do the data support the conclusions?

Reviewer #1: Partly

2. Has the statistical analysis been performed appropriately and rigorously? 

Reviewer #1: Yes

3. Have the authors made all data underlying the findings in their manuscript fully available?

Reviewer #1: Yes

4. Is the manuscript presented in an intelligible fashion and written in standard English?

Reviewer #1: Yes

5. Review Comments to the Author

Reviewer #1: Title does not look meaningful and attractive. It may be changed and result oriented.

Write up is very poor (view comment in attachment). Material and method section should be improved accordingly. Results do not support the title. Interpretation of results is not clear. Discussion section must be re-written before re-submission.

Conclusion section must be linked with the quantified results. General statements commonly mislead the readers.

Over all, manuscript is properly managed. English language may be improved. There are lot of language and grammar mistakes.

6. PLOS authors have the option to publish the peer review history of their article (what does this mean?). If published, this will include your full peer review and any attached files.

Reviewer #1: No

---

## [Author Response · Author response to Decision Letter 0]

18 May 2022

06 May 2022

We gratefully acknowledged the academic editor and reviewer for their valuable comments to our manuscript. The following are point-by-point responses to the academic editor and reviewer comments for the manuscript number: PONE-D-22-02551.

 Reviewer #1: 

Q#1: Title does not look meaningful and attractive. It may be changed and result oriented. 

Answer: Modified as ‘Knowledge, attitude and practice on household solid waste management in Gelemso town, Ethiopia’.

Q#2: Write up is very poor (view comment in attachment). 

Answer: We thank the reviewer for valuable comments. We have made significant improvement on the write up of the manuscript according to the comments. The revised parts are indicated in the track change form. 

NB: We confirm that there is no attached documents/comments with the editorial manager website/by email. Thus, we have done the correction by ourselves.

Q#3: Material and method section should be improved accordingly.

Answer: Again we appreciate the reviewer for valuable points. We had addressed the issue properly and indicated in track change form.

Q#4: Results do not support the title. Interpretation of results is not clear. 

Answer: The authors also acknowledged the reviewer for critical observation. Following the comments, we have corrected the points which are not clear for the readers, some points lacking the right interpretations are corrected. By now you will find it clear and legible. Again the corrections are indicated in revised manuscript with track change. 

Q#5: Discussion section must be re-written before re-submission.

Answer: We found points which are really required clarification, thus we have done the revisions properly. Plus to this, we made a general revision to the discussion section and all the mistakes and unclear parts are significantly improved and the changes are indicated by track change form.

Q6#. Conclusion section must be linked with the quantified results. General statements commonly mislead the readers.

Answer: Thank you very much for the comment raised by the reviewer. We have carefully corrected. The corrected points are indicated in the marked version with track change. 

Q7#. English language may be improved. There are lot of language and grammar mistakes.

Answer: Thank you very much for the comment raised by the reviewer. We have thoroughly corrected the English issues of the manuscript and significantly improved the manuscript content accordingly. The corrected points are indicated in the marked version with track change.

---

## [Decision Letter · Decision Letter 1]

7 Oct 2022

PONE-D-22-02551R1Knowledge, attitude and practice on household solid waste management in Gelemso town, EthiopiaPLOS ONE

Dear Dr. Yifat,

Thank you for submitting your manuscript to PLOS ONE. After careful consideration, we feel that it has merit but does not fully meet PLOS ONE’s publication criteria as it currently stands. Therefore, we invite you to submit a revised version of the manuscript that addresses the points raised during the review process. As explained previously, since the transfer in editorship this has been sent to a new reviewer, who has provided detailed comments for improvement.   I appreciate this is frustrating, but I am going to ask you to address their comments as well.

We look forward to receiving your revised manuscript.

Kind regards,

Alison Parker

Academic Editor

PLOS ONE

Reviewers' comments:

Reviewer's Responses to Questions

**Comments to the Author**

1. If the authors have adequately addressed your comments raised in a previous round of review and you feel that this manuscript is now acceptable for publication, you may indicate that here to bypass the “Comments to the Author” section, enter your conflict of interest statement in the “Confidential to Editor” section, and submit your "Accept" recommendation.

Reviewer #1: All comments have been addressed

Reviewer #2: (No Response)

2. Is the manuscript technically sound, and do the data support the conclusions?

Reviewer #1: Yes

Reviewer #2: Partly

3. Has the statistical analysis been performed appropriately and rigorously? 

Reviewer #1: Yes

Reviewer #2: I Don't Know

4. Have the authors made all data underlying the findings in their manuscript fully available?

Reviewer #1: Yes

Reviewer #2: No

5. Is the manuscript presented in an intelligible fashion and written in standard English?

Reviewer #1: Yes

Reviewer #2: Yes

6. Review Comments to the Author

Reviewer #1: Authors have addressed all the comments accordingly in the revised manuscript. Authors have addressed all the comments accordingly in the revised manuscript.

Reviewer #2: Dear authors,

This is my first revision of your article in which I ask for major revisions. Here are my comments:

1. Introduction

General comment:

I found the Introduction incomplete and not well structured. You fail to introduce the knowledge gap your article is coming to fill. When you focalize on Gelemso, the paragraph is disconnected from the first part. Moreover, you point to institutions as the main source of SW, but then you focus your research on households. Also, you do not include Khat consumption questions in your survey (or at least this is not presented in the Results analysis).

You said, “Although there are limited SWM facilities in the town, most of the time the solid waste have been indiscriminately disposed along roadsides and into open areas and hence the current status of the SWM problem in the town has reached a critical stage.” But both facts may be correlated (as your results show, by the way).

And then you state: “This shows that the people's overall understanding about solid waste and SWM practices in the town are not well established.” However, this is a hypothesis and not a statement.

You should restructure the Introduction to better explain why the information you present in the article is important and what knowledge gap are you filling.

Specific comments:

Two first phrases of the 2nd paragraph should be unified.

2. Methods

Specific comments:

In section 2.1, a description of the waste management system of Gelemso is needed. What kind of facilities people have access to, to dispose waste? Waste pickers are active in Gelemso? Have the Municipality a public waste collection service? Of what kind? How many containers are available? Who pick up waste from containers and where it is waste finally disposed? All these questions are crucial to understand the answers to the survey questions.

In section 2.2, You should explain from which list you made the systematic random sampling. Do you have a list of all the households? Selected households they all accepted to participate in the study? A satellite image of the town with the identification of the selected households would be a plus.

In 2.3, the structure of the paragraph describing definitions should be reformulated because it is not understandable.

You skip from 2.4 to 2.7

3. Results (General comment: all the data should be available)

Section 3.1: information presented in this section is not used in the Discussion section. Either you use it in section 4, or you remove it from section 3.

Section 3.2: You say, “About 96% of the households gave the correct response regarding solid waste as the major pollutant of the environment”. Why can you affirm that SW is the main environmental pollution source?

Many questions are not well formulated:

Does solid waste pollute the environment?  It depends on how it is managed

Do you know solid waste is being a wealth?  You give the answer in the question. "Do you consider..." "Do you think" "Do you perceive..."

The amount of solid waste can be reduced by reusing at household level?  It is an obvious question

Sorting of solid waste at home level helps for SWM?  It depends on how the system is structured and you did not gave the reader any elements to know that.

Section 3.3: You give % of waste composition. But waste characterization studies have specific methodologies, and it seems you do not follow any of them. You should remove Figure 1 and all mentions of % of types of waste from your manuscript.

You say, “However, the majority (79.5%) of the households disposed of their solid waste illegally, either in the backyard with sacs or along the roadsides and in gully (Table 4).” But may be they expected a municipal kerbside collection. That’s why the Solid Waste Management system description is so important.

You say: “The majority (89.2%) of the households reported the absence of adequate solid waste landfills or damping sites in the city and thus had practiced an improper SWM (Table 4)”. � It seems the problem is not people are not well informed but the absence of infrastructure!

3.4 In the questions you included: “How do you get rid of solid wastes from home”. Do you considered the possibility of people giving different destinations to recycling waste and non-recycling waste?

4. Discussion

First paragraph can be part of the Introduction.

After reading the discussion, it is still not clear why improper disposal is occurring in Gelemso.

5. Conclusion

Should be modified accordingly.

7. PLOS authors have the option to publish the peer review history of their article (what does this mean?). If published, this will include your full peer review and any attached files.

Reviewer #1: **Yes: **Shahbaz Khan

Reviewer #2: No

---

## [Author Response · Author response to Decision Letter 1]

27 Oct 2022

Date: 25 October 2022

We are very much grateful to the academic editor who managed this manuscript again and also for the reviewers for their valuable comments to our manuscript. The following are point-by-point responses to reviewer comments for the manuscript number: PONE-D-22-02551R1.

 Reviewer #2: 

First of all, I would like to thank the second reviewer, who reviewed this manuscript and provided us the most valuable comments and suggestions. We received all the comments positively and frankly specking the manuscript is now well enriched. 

Q#1: Introduction

General comment:

I found the Introduction incomplete and not well structured. You fail to introduce the knowledge gap your article is coming to fill. 

Answer: The whole introduction section is now modified according to the comments.

When you focalize on Gelemso, the paragraph is disconnected from the first part. Moreover, you point to institutions as the main source of SW, but then you focus your research on households. Also, you do not include Khat consumption questions in your survey (or at least this is not presented in the Results analysis).

Answer: The specific paragraph as well as the focus area of the research has been modified following the comments.

You said, “Although there are limited SWM facilities in the town, most of the time the solid waste have been indiscriminately disposed along roadsides and into open areas and hence the current status of the SWM problem in the town has reached a critical stage.” But both facts may be correlated (as your results show, by the way). And then you state: “This shows that the people's overall understanding about solid waste and SWM practices in the town are not well established.” However, this is a hypothesis and not a statement.

Answer: We accept the comments and modified the sentence accordingly.

Specific comments:

Two first phrases of the 2nd paragraph should be unified.

Answer: The comment is corrected

Q#2: Methods

Specific comments:

In section 2.1, a description of the waste management system of Gelemso is needed. What kind of facilities people have access to, to dispose waste? Waste pickers are active in Gelemso? Have the Municipality a public waste collection service? Of what kind? How many containers are available? Who pick up waste from containers and where it is waste finally disposed? 

Answer: A paragraph explaining the whole waste management system of the Gelemso municipality including kind of facilities, waste pickers, waste collectors, the presence and number of shared container, damping sites and how the collected wastes are finally damped to the damping site are included in the “Method section” under sub section 2.1. 

In section 2.2, You should explain from which list you made the systematic random sampling. Do you have a list of all the households? Selected households they all accepted to participate in the study? 

Answer: The whole lists of the households were obtained from municipality office, and the section was made from the lists. The numbers of actual household who completed the questionnaire from the total sample size (396) were 390 (98.5% response rate). The following comments are now included and corrected in the revised manuscript.

A satellite image of the town with the identification of the selected households would be a plus.

Answer: We have included map of study area in supplementary file.

In 2.3, the structure of the paragraph describing definitions should be reformulated because it is not understandable.

Answer: Corrected as you suggested

You skip from 2.4 to 2.7

Answer: Corrected as you suggested

Q#3: Results (General comment: all the data should be available)

Section 3.1: information presented in this section is not used in the Discussion section. Either you use it in section 4, or you remove it from section 3.

Answer: Now included in the discussion part.

Section 3.2: You say, “About 96% of the households gave the correct response regarding solid waste as the major pollutant of the environment”. Why can you affirm that SW is the main environmental pollution source?

Answer: Now modified in the revised version.

Many questions are not well formulated:

Does solid waste pollute the environment?  It depends on how it is managed

Do you know solid waste is being a wealth?  You give the answer in the question. "Do you consider..." "Do you think" "Do you perceive..." 

The amount of solid waste can be reduced by reusing at household level?  It is an obvious question

Sorting of solid waste at home level helps for SWM?  It depends on how the system is structured and you did not gave the reader any elements to know that.

Answer: Some of the questions including the mentioned questions were modified in revised version.

Section 3.3: You give % of waste composition. But waste characterization studies have specific methodologies, and it seems you do not follow any of them. You should remove Figure 1 and all mentions of % of types of waste from your manuscript.

Answer: All types of waste composition results were removed from the manuscript.

You say, “However, the majority (79.5%) of the households disposed of their solid waste illegally, either in the backyard with sacs or along the roadsides and in gully (Table 4).” But maybe they expected a municipal kerbside collection. That’s why the Solid Waste Management system description is so important.

You say: “The majority (89.2%) of the households reported the absence of adequate solid waste landfills or damping sites in the city and thus had practiced an improper SWM (Table 4)”. It seems the problem is not people are not well informed but the absence of infrastructure!

Answer: Descriptions of the solid waste management system in Gelemso town were described briefly in the revised manuscript. I hope the question is now understandable for the readers.

3.4 In the questions you included: “How do you get rid of solid wastes from home”. Do you consider the possibility of people giving different destinations to recycling waste and non-recycling waste?

Answer: Yes.

Q#4: Discussion

First paragraph can be part of the Introduction.

Answer: The paragraph was deleted from the discussion part according to the suggestion.

After reading the discussion, it is still not clear why improper disposal is occurring in Gelemso.

Answer: Now the discussion part is modified to why improper sold waste disposal is occurred in the study site.

Q#5: Conclusion: Should be modified accordingly.

Answer: The conclusion part has modified according to the comment in the revised manuscript.

---

## [Decision Letter · Decision Letter 2]

14 Nov 2022

Knowledge, attitudes and practices on household solid waste management and associated factors in Gelemso town, Ethiopia

PONE-D-22-02551R2

Dear Dr. Yifat,

We’re pleased to inform you that your manuscript has been judged scientifically suitable for publication and will be formally accepted for publication once it meets all outstanding technical requirements.

Kind regards,

Alison Parker

Academic Editor

PLOS ONE

Additional Editor Comments (optional):

Reviewers' comments:

Reviewer's Responses to Questions

**Comments to the Author**

1. If the authors have adequately addressed your comments raised in a previous round of review and you feel that this manuscript is now acceptable for publication, you may indicate that here to bypass the “Comments to the Author” section, enter your conflict of interest statement in the “Confidential to Editor” section, and submit your "Accept" recommendation.

Reviewer #2: All comments have been addressed

2. Is the manuscript technically sound, and do the data support the conclusions?

Reviewer #2: Yes

3. Has the statistical analysis been performed appropriately and rigorously? 

Reviewer #2: Yes

4. Have the authors made all data underlying the findings in their manuscript fully available?

Reviewer #2: Yes

5. Is the manuscript presented in an intelligible fashion and written in standard English?

Reviewer #2: Yes

6. Review Comments to the Author

Reviewer #2: Congratulations for your job improving the article. You addressed all the comments and now the article is much better.

7. PLOS authors have the option to publish the peer review history of their article (what does this mean?). If published, this will include your full peer review and any attached files.

Reviewer #2: No

---

## [Editor Report · Acceptance letter]

13 Jan 2023

PONE-D-22-02551R2 

Knowledge, attitudes and practices on household solid waste management and associated factors in Gelemso town, Ethiopia 

Dear Dr. Tigu:

I'm pleased to inform you that your manuscript has been deemed suitable for publication in PLOS ONE. Congratulations! Your manuscript is now with our production department. 

Kind regards, 

on behalf of

Dr. Alison Parker 

Academic Editor

PLOS ONE